# Endometrial Cancer: 2023 Revised FIGO Staging System and the Role of Imaging

**DOI:** 10.3390/cancers16101869

**Published:** 2024-05-14

**Authors:** Manuel Menendez-Santos, Carlos Gonzalez-Baerga, Daoud Taher, Rebecca Waters, Mayur Virarkar, Priya Bhosale

**Affiliations:** 1Department of Radiology, University of Florida College of Medicine-Jacksonville, Jacksonville, FL 32209, USA; carlos.gonzalezbaerga@jax.ufl.edu (C.G.-B.); mayur.virarkar@jax.ufl.edu (M.V.); 2Department of Radiology, University of Texas MD Anderson Cancer Center, Houston, TX 77030, USA; daoud.taher@mdanderson.org (D.T.); rebecca.waters@mdanderson.org (R.W.); priya.bhosale@mdanderson.org (P.B.)

**Keywords:** endometrial cancer, MRI, FIGO, molecular

## Abstract

**Simple Summary:**

Emerging endometrial cancer research allows for a better understanding of the natural progression of this disease. This, in turn, gives way to better strategies by which to identify and effectively treat endometrial cancer. The International Federation of Gynecology and Obstetrics (FIGO) uses this compendium of evolving research to establish and subsequently update a staging classification system for endometrial cancer. Imaging studies are crucial in the diagnosis of endometrial cancer, and with the most recent update to the FIGO endometrial cancer staging system, their role must be reviewed.

**Abstract:**

The FIGO endometrial cancer staging system recently released updated guidance based on clinical evidence gathered after the previous version was published in 2009. Different imaging modalities are beneficial across various stages of endometrial cancer (EC) management. Additionally, ongoing research studies are aimed at improving imaging in EC. Gynecological cancer is a crucial element in the practice of a body radiologist. With a new staging system in place, it is important to address the role of radiology in the EC diagnostic pathway. This article is a comprehensive review of the changes made to the FIGO endometrial cancer staging system and the impact of imaging in the staging of this disease.

## 1. Introduction

Endometrial cancer (EC) is the second most prevalent cancer among women [1]. Despite advances in cancer research, endometrial cancer incidence and mortality are worsening [2]. Uterine corpus malignancies have increased by 1% per year in females over 50 since the mid 2000s and 2% in younger women since the mid 1990s. By the end of 2023, there will be an estimated 66,200 new cases and 13,030 related deaths in the United States [3].

To focus efforts on up-to-date diagnosis, staging, and management, the International Federation of Gynecological and Obstetrics (FIGO) established a staging system that is gradually revised based on the latest clinical evidence. The FIGO 2023 staging system (revised from FIGO 2009) integrates advances in the pathologic and molecular understanding of endometrial cancer progression [4]. Imaging is not directly included in the staging criteria but is essential in guiding management alternatives for patients diagnosed with EC. In this article, we review and compare the previous and updated FIGO endometrial cancer staging systems and discuss the essential role that imaging plays in the latest revised system.

## 2. Histopathology

Endometrial cancer subtypes are categorized according to histopathology. In 2020, the WHO published its 5th edition of the classification of female genital tumors. The classification of subtypes is a significant predictor of prognosis and is currently involved in the revised FIGO 2023 staging. Non-aggressive histological types include grade 1 and 2 endometrial endometrioid carcinomas (EEC), the most common type [5]. Aggressive histological types include high-grade (grade 3) EEC, serous carcinoma (SC), clear cell carcinoma (CCC), mixed carcinoma (MC), undifferentiated carcinoma (UC), carcinosarcoma (CS), gastro-intestinal type and mesonephric-like mucinous carcinomas (Figure 1) [6,7].

EEC is the most common histologic subtype, accounting for approximately 85% of known cases [7,8]. Low-grade EEC includes grade 1 and grade 2 EEC. Grade 1 EECs demonstrate ≤ 5% solid, non-glandular composition. Grade 2 EECs show a 5–50% solid, non-glandular component. Lesions with ≥ 50% solid component are classified as high-grade (Grade 3) EEC [4,9]. High-grade EECs are more complex, and studies have shown that further classification of this subtype guides prognosis and management [8,10].

## 3. Molecular Classification

In 2013, the Cancer Genome Atlas (TCGA) identified four molecular subtypes of high-grade EEC, POLE_mut_, MMR_d_, P53_abn_, and no specific molecular profile (NSMP), depending on the genetic architecture [2,11]. This study showed that molecular classification was more accurate than histological classification alone in EEC, given their morphologic heterogeneity [10]. This novel study was the foundation for further studies demonstrating prognostic data based on molecular subtypes, ultimately included in the revised FIGO staging (Figure 2).

POLE_mut_ has been shown to have a more favorable prognosis and is more frequently expressed in younger patients with endometrial cancer [13]. P53_abn_ has been significantly associated with the lowest recurrence-free survival of all of the molecular subtypes [14]. The prognostic significance of either MMR_d_ or NSMP is still unclear and, therefore, has no impact on current FIGO staging. MMR_d_ mismatch mutations should, however, prompt further workup of Lynch syndrome when clinically appropriate [15]. This clinical significance in POLE_mut_ and P53_abn_ subgroups now correlates with a lower or higher FIGO staging.

## 4. FIGO 2023 Staging

The latest revision of the FIGO staging includes histopathologic and molecular criteria for the classification and sub-staging of lesions based upon recent data from ESGO/ESTRO/ESP guidelines and WHO criteria [4]. This provides more accurate definitions of prognostic groups with which to direct appropriate disease management (Table 1). The criteria for staging endometrial cancer were tumor grade, myometrial invasion, lymphovascular space invasion (LVSI), cervical stroma invasion, adnexal involvement, lymph node status, and molecular classification (Figure 3).

### 4.1. Stage I

In the FIGO 2009 revision, stage I lesions were limited to the body of the uterus with a depth of invasion of less or greater than 50%, IA and IB, respectively.

In the 2023 revision, histologic aggression of the endometrial carcinoma differentiates the classification. Table 2 compares the 2009 and 2023 FIGO revisions for stage 1. Non-aggressive histologic tumors are composed of low-grade endometrioid carcinoma (ECC). At the same time, more aggressive types are made up of serous carcinoma (SC), clear cell carcinoma (CCC), mixed carcinoma (MC), undifferentiated carcinoma (UC), carcinosarcoma (CS), mesonephric-like and gastrointestinal mucinous type carcinomas [16].

Low-grade EC is classified as stage I. However, it can be further subclassified based on the involvement of the myometrium and lymphovascular space (LVSI). Lymphovascular space involvement is categorized as none and focal (<5 vessels). A lesion limited to a polyp or within the endometrium is considered Stage IA1. Less than 50% of the myometrium, with no focal LVSI, is stage IA2 (Figure 4). Stage IA3 was added and is used when low-grade ECC is limited to the uterus and ovary but without invasion of the ovarian capsule. Greater than 50% of myometrial involvement by a low-grade tumor is staged as IB (Figure 5 and Figure 6). Any histologically aggressive tumor or high-grade ECCs confined to the endometrium or as a polyp is given a stage IC classification (Figure 7).

### 4.2. Stage II

The FIGO 2009 classification determined that stage II has cervical stroma invasion without extension beyond the uterus. The newly revised classification determined that invasion at the level of, or deeper than, a benign endocervical crypt with extrauterine extension is stage IIA (Figure 8), substantial LVSI is considered stage IIB, while invasion of histologically aggressive tumors involving the myometrium is considered stage IIC (Figure 9). Table 3 highlights the differences between FIGO 2009 and FIGO 2023.

### 4.3. Stage III

For FIGO 2009, invasion of structures outside the uterus without spreading to the inner lining of the rectum or bladder is categorized as stage III. The invasion of serosa of the uterus, adnexa, or both was considered IIIA (Figure 10), and parametrial extension was IIIB (Figure 11). Stage IIIC lesions were subdivided depending on pelvic node involvement (IIIC1) and para-aortic lymph node involvement (IIIC2) (Figure 12).

With the new 2023 revision, Stage III is subdivided based on the spread of the tumor within the pelvis. The invasion of serosa of the uterus, adnexa, or both is classified as IIIA. Stage IIIB now includes vaginal, parametrial, or pelvic peritoneal involvement. Stage IIIC is subdivided depending on pelvic node involvement (IIIC1) and para-aortic lymph node involvement (IIIC2).

Stage IIIC can be further subclassified based on micro- and macrometastasis. Micrometastasis is defined as lesions that are 0.2–2 mm in size and/or more than 200 cells, while macrometastasis is larger than 2 mm. Micrometastasis is indicated as an “i” notation, and macrometastasis is marked as an “ii” notation. For example, a patient’s lesion involving micrometastasis to pelvic lymph nodes is categorized as IIIC1i. These differences are shown in Table 4.

### 4.4. Stage IV

In FIGO 2009, tumor invasion of organs outside the uterus or imaging signs of metastasis or lesions are classified as stage IVA or IVB. The recent FIGO staging provides an additional substage. Stage IVA is bladder mucosal or intestinal/bowel mucosal invasion (Figure 13), while abdominal peritoneal metastasis beyond the pelvis is stage IVB (Figure 14 and Figure 15). Lastly, stage IVC is used when distant metastases are identified (Figure 16). These updates are highlighted in Table 5.

### 4.5. FIGO Staging with Molecular Classification

The Cancer Genome Atlas classified endometrial cancer into four classes: POLE/ultramutated, MMR_d_ microsatellite instability, somatic copy number alteration high/serous like (SCNA-high), and somatic copy number alteration low (SCNA-low). Using a surrogate that includes three markers and one molecular test, an analysis for pathogenic POLE mutations classifies endometrial cancer into four groups: POLE_mut_, MMR_d_, p53_abn_, and NSMP. POLE_mut_ indicates a favorable prognosis, while p53_abn_ has a worse prognosis. MMR_d_ and NSMP are intermediate in prognosis.

Based on clinicopathological and imaging features, endometrial cancer patients can be staged and provide a prognosis. Based on this, their molecular classification determines surgical or medical treatments. Therefore, stages III and IV with a molecular classification can have an “m” annotation with the appropriate molecular class.

## 5. Role of Imaging Modalities

### 5.1. Ultrasound (US)

The most common initial symptom of endometrial cancer is abnormal uterine bleeding (AUB) [17]. The standard initial workup for AUB includes a transvaginal ultrasound (TVUS) [18,19]. The ACR appropriateness criteria for abnormal uterine bleeding classifies several TVUS, transabdominal pelvic ultrasounds, and US duplex Doppler pelvic ultrasounds as adequate for initial imaging of AUB [20]. US techniques are usually combined for better assessment of pelvic structures. Several studies have revealed that measuring ≤ 4 mm endometrial lining correlates to ≥ 99% negative predictive value for endometrial cancer in patients presenting with AUB [21,22,23,24,25] (Figure 17). Three-dimensional TVUS (3D-TVUS) is an emerging US technique with good diagnostic potential. A comparison study between 3D-TVUS and 2D-TVUS showed variable results but no statistical significance in sensitivity and specificity [26]. In a recent metanalysis comparing 3D-TVUS with MRI for diagnosis of deep myometrial invasion of EC (>50% EC involvement of myometrium), 3D-TVUS was shown to have a pooled sensitivity of 77%, compared with 80% for MRI [27]. When compared with MRI for the assessment of cervical involvement, Spagnol et al. concluded that 3D-TVUS has respective sensitivity and specificity of 75% and 86% vs. 83% and 82% for MRI. However, the differences were not statistically significant [27]. Additionally, Green et al. reported dynamic contrast-enhanced US (DCE-US) as more sensitive to detecting myometrial invasion and cervical stromal invasion, displaying a sensitivity of 74% and 75%, respectively [28]. These studies show that emerging US technologies may offer promising value in diagnosing endometrial cancer. Another limitation is operator dependence, suggesting that results may vary in the hands of experts. Overall, US provides lower diagnostic value for staging than other imaging modalities but functions as an efficient first-line screening tool, and early detection is paramount. FIGO 2023 staging highlights the importance of earlier diagnosis for improved prognosis. Additionally, early detection may lead to earlier genetic analysis of the lesion via biopsy, which is now needed for staging.

### 5.2. Computed Tomography (CT)

Computed tomography (CT) imaging is widely used to assess systemic diseases, as it is readily available and offers rapid image acquisition. Its role pertains to the workup and surveillance of advanced endometrial cancer. CT imaging of the chest, abdomen, and pelvis with intravenous contrast is appropriate for providing information about distant metastatic disease and lymph node involvement [12,29,30]. New 2023 FIGO updates to stage IV endometrial cancer may further increase the utility of CT imaging in advanced disease due to its ability to identify peritoneal metastasis and distant metastatic disease (Figure 16). Mazzei et al. have determined that the sensitivity, specificity, PPV, NPV, and accuracy of detecting peritoneal carcinomatosis of primary ovarian cancer using MDCT was 100%, 40%, 93%, 100%, and 93%, respectively [31]. CT is comparable with laparoscopy in the accuracy of detecting peritoneal disease with 94.9% sensitivity, 86.7% specificity, 97.9% positive predictive value (PPV), 72.2% negative predictive value (NPV), and 93.8% accuracy [32]. Hauge et al. proposed contrast-enhanced CT texture analysis as a prognostic indicator of EC, but larger scale studies are needed to further evaluate this CT application field [31]. CT imaging of localized disease is inferior to other imaging modalities due to the lower soft tissue resolution [12,29,30,33]. Other disadvantages of this modality include contrast infusion contraindications in patients with renal disease and radiation exposure.

### 5.3. Magnetic Resonance Imaging (MRI)

Since the publication of the 2009 FIGO staging guidelines, the benefit of MRI for endometrial cancer diagnosis and staging has been well studied [12,14,29,30,34,35]. The high degree of soft tissue resolution allows for a superior lesion assessment and contributes to better guided treatment in these patients. Table 6 highlights pelvic MRI parameters for gynecologic cancer workup. Figure 18 highlights MRI’s high degree of imaging resolution in normal female pelvic anatomy. Pelvic MRI, as part of the initial workup, assists in identifying the cellular origin of the tumor (endometrial vs. endocervical), local extent, presence and extent of myometrial invasion, and lymph node involvement [34,36,37].

In terms of technique, using axial oblique MRI angled perpendicularly to the endometrial cavity allows for a better assessment of the myometrial invasion [35]. For optimal image resolution and quality, a 1.5-Tesla or 3-Tesla magnet is recommended, along with a multi-parametric combination of T2-weighted imaging (T2WI), diffusion-weighted imaging (DWI), and dynamic contrast imaging (DCEI). Hori et al. have demonstrated that a 3.0 T improved image quality compared with a 1.5 T, likely due to higher magnetic fields, resulted in an improved signal-to-noise ratio (29). Glucagon is spasmolytic to decrease bowel peristalsis, reduce motion artifact, and improve image quality. The use of saturation bands is another way of reducing motion artifacts. Vaginal distension is not mandatory but may aid in detecting suspected extra-uterine invasion. This can be done with the use of vaginal gel, and allows for better visualization of the pelvic floor [35,38].

On T2WI, routine endometrial imaging is observed as the high signal intensity of endometrial tissue surrounded by the low signal intensity of the junctional zone, further surrounded by the outer myometrium of intermediate signal intensity. Endometrial lesions may display a heterogeneous signal intensity but are most commonly hyperintense compared with the myometrium. This signal intensity distinction on T2WI of the lesion from normal myometrium helps assess for the presence of myometrial invasion, a crucial determinant of tumor staging. Some lesions may not show distinct delineation on T2WI. [39], which presents a challenge in determining the extent of myometrial invasion. DCEI and DWI provide additional layers of information to increase diagnostic accuracy.

Early phase DCEI (30–60 s after contrast infusion) shows enhancement of sub-endometrial lining. Thus, observing the uninterrupted hyper-enhanced lining in the early phase may exclude myometrial invasion. Deep myometrial invasion can be assessed during the equilibrium phase (120–180 s after contrast infusion). During this phase, there is maximal hyper-enhancement of the myometrium, allowing better characterization of malignant tissue. In the delayed phase (4–5 min after contrast infusion), cervical stromal invasion can be evaluated [12,40]. Some endometrial tumors may present as poorly delineated on T2 imaging, hindering the accurate assessment of tumor extent. Large tumors, for example, can expand the endometrial cavity, compressing and distorting the surrounding anatomy. Additionally, coexisting uterine pathologies, such as adenomyosis and fibromas, may alter the uterine anatomy and interfere with the evaluation of endometrial tumor extent. DWI can provide a more accurate visualization of tumor borders in these cases. Lesions on DWI are observed as hyperintense on high *b*-value series (500–1000 s/mm^2^) and show a low apparent diffusion coefficient (ADC) [12,34,35,40]. MRI can detect metastatic lymph nodes as small as 5 mm if they show restricted diffusion and low ADC [40]. Rechichi et al. have reported that endometrial carcinoma may be differentiated from normal tissue with an ADC value of less than 1. However, there was no significant difference in ADC values of endometrial cancer tissue in tumor grade, depth of myometrial invasion, or presence of lymph node metastasis [41]. Ongoing research aims to correlate ADC values and tumor volumes with the EC staging [42,43]. DWI is also helpful in detecting vaginal metastasis. This can be seen as a direct tumor spread into vaginal tissue or separate implants [35]. Accurate information on tumor involvement of surrounding structures has a higher impact on endometrial cancer staging. This is due to special considerations of tumor histopathological and molecular information. Previously, stage I was subdivided into stage IA if <50% of myometrium involvement was observed and stage IIB if >50% of myometrial involvement was observed. If any myometrial involvement is marked with a tumor with aggressive histopathology, FIGO 2023 now upstages to stage IIC.

Cervical stromal invasion is also a strong determinant of endometrial cancer staging. Detection of cervical stromal invasion of a non-aggressive histological type is the definition of stage IIA EC. MRI is crucial in the presurgical phase of treatment to determine cervical stromal invasion. Disruption of the low signal intensity of the cervical stroma on T2WI suggests tumor invasion. Other diagnostic criteria include disruption of the normal enhancement of stromal tissue by the hypo-enhancing tumor on DCE-MRI and high signal intensity of the cervical area on a high *b*-value DWI [12]. Two single-center retrospective studies aimed at determining the accuracy of MRI for cervical stromal invasion reported 93.2% and 98.9% accuracy, respectively [36,37]. A large-scale multicenter retrospective study concluded that MRI accuracy was 89.3% when detecting cervical stromal invasion [44]. The accuracy of MRI in detecting cervical stromal invasion is mainly due to high specificity, as it has been reported to have low sensitivity in multiple studies [36,37,45]. A common limitation reported in these studies is the detection of the microscopic invasion of cervical tissue, which is challenging in MRI imaging.

When evaluating for peritoneal metastasis with MRI imaging, the diagnostic accuracy depends on the lesion size. For peritoneal lesions > 10 mm, MRI offers a similar sensitivity of approximately 90–95%. However, for smaller lesions between 5–10 mm, MRI is superior [46]. Size is also important when assessing lymph node involvement. Identification of pelvic lymph nodes with a short axis > 8 mm or abdominal lymph nodes > 10 mm should be noted as suspicious for metastasis, regardless of signal intensity or borders. Another sign of suspected lymph node involvement is lymph node clustering near the tumor. A limitation of these assessments is that micrometastasis may be present in a normal-sized lymph node [35]. Ferumoxytol, an FDA-approved MR lymphography agent, may detect metastasis in lymph nodes independent of its size. However, it has only been used for prostate, bladder, and kidney cancer [47].

### 5.4. Positron Emission Tomography/Computed Tomography (PET/CT)

Along with CT, positron emission tomography/computed tomography (PET/CT) adds to the imaging options for the adequate assessment of advanced endometrial cancer. Several large-scale studies have been published in the last decade, defining PET/CT value in pre- and post-op/surveillance phases. PET/CT’s role in preoperative staging is that of detecting distant metastasis.

Peritoneal implants, for example, manifest as focal or diffuse abnormal 18F-Fluorodeoxyglucose (FGD) uptake in bowel serosa, peritoneum, or omentum [46] (Figure 19). Reported sensitivity is 58–100%, due to a high false negative rate. This is primarily because physiologic, metabolic activity in the bowel loops obscures the serosal bowel implants [48].

Research endeavors have also focused on assessing the accuracy of PET/CT in lymph node involvement. A multicentric, French retrospective study using PET/CT with FDG concluded 61.8% sensitivity and 86.1% specificity in detecting para-aortic lymph node disease [49]. A meta-analysis to determine FDG-PET/CT accuracy in diagnosing recurrent disease in endometrial cancer patients concluded with 95.8% sensitivity and 92.5% specificity for this modality [50]. Fasmer et al. demonstrated that PET/CT has better detection of lymph node metastasis in high-risk patients than MRI, with a sensitivity and specificity of 56% and 90%, respectively [51]. However, a single-centered prospective study by Stewart et al. aimed to determine PET/CT accuracy and determined that the high false negative rate (54.2%) proves that PET/CT alone should not be used to rule out lymph node metastasis.

Additionally, surgical lymph node staging remains to be superior to imaging modalities [52]. It is essential to clarify that PET/CT is limited to the detection of lymph node macrometastasis instead of micrometastasis. Micrometastasis is defined as 0.2 mm–2 mm in size and is mainly a pathological finding of extracted local lymph nodes in the surgery and sentinel lymph node biopsy assessment intraoperatively [4]. Despite its limitations, PET/CT remains essential in the preoperative evaluation of distant metastatic disease.

### 5.5. Positron Emission Tomography/Magnetic Resonance Imaging (PET/MRI)

PET/MRI is an emergent modality that aims to combine the high-resolution anatomic differentiation of MRI with the functional, metabolic assessment of PET (Figure 19). A study by Tusyoshi et al. compared PET/MRI with contrast-enhanced MRI (ceMRI) and determined that the statistically significant lesion-based sensitivity, specificity, and accuracy for regional nodal metastasis were 100, 96.9, and 97%, respectively [53]. In a recent prospective cohort study, PET/MRI achieved an accuracy of 77% and 91% for correctly identifying myometrial and lymph node involvement, respectively. PET/MRI parameters that predict myometrial and lymph node involvement include total lesion glycolysis, volume index, and SUV_max_/ADC_mean_ ratio, for which volume index was the most sensitive metric [54]. Additionally, PET/MRI has been evaluated to assess the extent of peritoneal disease at primary diagnosis, in order to determine surgical feasibility in patients with advanced disease [55]. These studies justify larger scale investigations to determine the uses of this emerging modality.

## 6. Treatment

Endometrial cancers (EC) can be treated with medical therapies, surgery, or radiotherapy based on staging (Figure 20) [56]. The National Comprehensive Cancer Network’s (NCCN) latest guidelines have determined the treatment based on three categories: (1) confinement to the uterus, (2) presence or possible cervical involvement, and (3) suspected extrauterine disease [57]. Surgical staging is more specific than other diagnostic strategies for determining myometrial invasion in EC. However, preoperative staging via imaging allows for more accurate surgical planning [40,58]. Stage 1 and 2 EC have excellent prognoses, while the lymphovascular invasion of grade 2 or 3 tumors yields low survival rates. Low-grade tumors confined to the uterus, grade 1 EC, can undergo total hysterectomy with or without bilateral salpingo-oophorectomy. Laparoscopic and robotic hysterectomies are preferred procedures [59,60,61]. Hormonal therapy is considered for those who desire fertility preservation or who have stage IA disease [57]. With the addition of molecular classification within the FIGO system, endometrial biopsies can provide more information and guide management.

Endometrial cancers with low-grade stage IA MMRd/NSMP and stage I-II POLEmut are not recommended for adjuvant treatment. Nevertheless, those with stage IB MMRd/NSMP, high-grade stage IA, and stage IA p53abn may benefit from vaginal brachytherapy or EBRT [62]. Stage IA clear cell or serous histology may benefit from pelvic radiation therapy alone or chemotherapy with or without vaginal brachytherapy [63,64]. Platinum and taxane-based adjuvant chemotherapy are recommended for stage I or II with high-risk features [62,65,66,67,68,69]. Using radiotherapy in the early stage is controversial. Nevertheless, it is commonly considered for late-stage (II or III) cases [70]. Preoperative radiotherapy can also be used for tumor debulking. For inoperable tumors, radiation alone has provided results comparable to those of patients who have undergone surgical management [71,72].

Stage III and IV lesions involve pelvic and para-aortic lymph node metastasis, which results in a less favorable prognosis for the patient. In a study by Kumar et al., the prevalence of pelvic and para-aortic lymph node metastasis was 17% and 12%, respectively. Lymphadenectomies are additional procedures that have shown improved overall and progression-free survival in high-risk patients and allow for clinical staging completion [61,73,74].

Imaging can potentially evaluate lymph node metastasis in these patients. MRI has a sensitivity of 44% and a specificity of 98% in detecting lymph nodes. Therefore, new alternatives are required to assess these patients. The latest modalities showing promising results are diffusion-weighted imaging, PET/CT, and MRI with ultra-small supermagnetic iron oxide [75].

Patients with advanced or recurrent disease are treated palliatively with carboplatin and paclitaxel as standard care [62,63,64,65,66]. However, hormonal therapy, progestins, tamoxifen, and medroxyprogesterone have been used and tolerated by patients. With the new molecular classification integrated into the FIGO staging, patients benefit from targeted chemotherapy, such as that involving checkpoint inhibitors, that is more effective against microsatellite unstable tumors, yet toxicity and tolerability are of concern [76,77,78].

Preoperative imaging allows multidisciplinary teams to guide treatment strategies, surgical planning, cost, and time management before surgery. Furthermore, it will empower patients to participate in the decision-making process regarding their treatment. Imaging can facilitate the assessment of myometrial invasion of EC and confinement to endometrium via DCE-MRI. This benefits younger patients by allowing fertility preservation, avoiding surgery, and beginning progestin therapy if the tumor is characterized as low-grade, such as those of stage IA [40,58,79,80]. In elderly patients, imaging may reduce morbidity and mortality rates by detecting lymph node metastases and precluding surgery if lymph node metastases are not evident on imaging. Due to the possible adverse effects of hormonal therapy, risks, and benefits should be weighed by patients and their healthcare providers [81].

## 7. Conclusions

Revising staging guidelines in any disease aims to improve patient outcomes and prognosis. Adopting the updated FIGO 2023 endometrial cancer staging system within clinical settings will result in a more comprehensive preoperative assessment of women with endometrial cancer. Consequently, this will allow for a more accurate analysis facilitating customized treatment strategies. A multimodal approach is beneficial to accurately stage the disease.

## Figures and Tables

**Figure 1 cancers-16-01869-f001:**
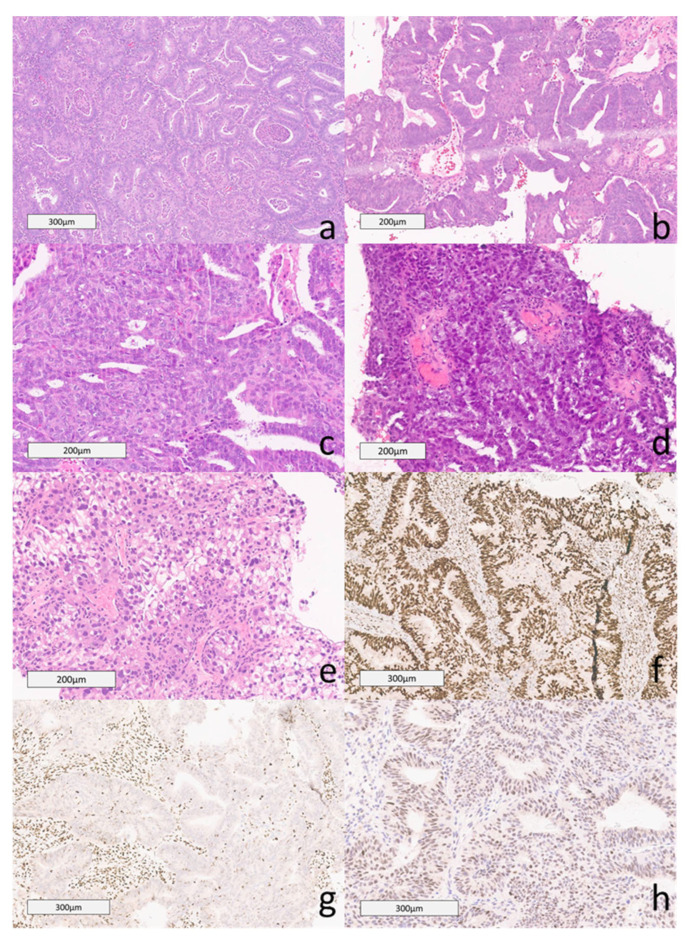
Histopathology of Endometrial Cancer. (**a**) Grade 1: The image shows irregular glands lined by columnar epithelium with pseudostratified nuclei and mild cytologic atypia. The tumor contained less than 5% of solid areas and was classified as Grade 1 (H&E). (**b**) Grade 2: These tumors contain 6% to 50% of solid components. This case contained approximately 40% solid areas, excluding foci of squamous differentiation (H&E). (**c**) Grade 3: This case showed a solid growth pattern in 80% of the tumor. The remainder was composed of glandular components (H&E). (**d**) Serous carcinoma in the omentum: This case of serous carcinoma was invading the omentum. The image shows papillary clusters of high-grade neoplastic cells (H&E). (**e**) Clear cell carcinoma comprises clear cells in papillary and solid growth patterns. The cytoplasm ranges from clear to eosinophilic. There is mild to marked cytologic atypia (H&E). (**f**) Intact MMR MSH6: Immunohistochemical stain of MSH6 with intact nuclear expression. (**g**) MSH6 Loss FIGO 2: Immunohistochemical stain of MSH6 with loss of nuclear expression within tumor cells. (**h**) P53 WILD type in FIGO 2 H&E: Endometrial adenocarcinoma, FIGO grade 2, had p53 wildtype pattern of expression (H&E).

**Figure 2 cancers-16-01869-f002:**
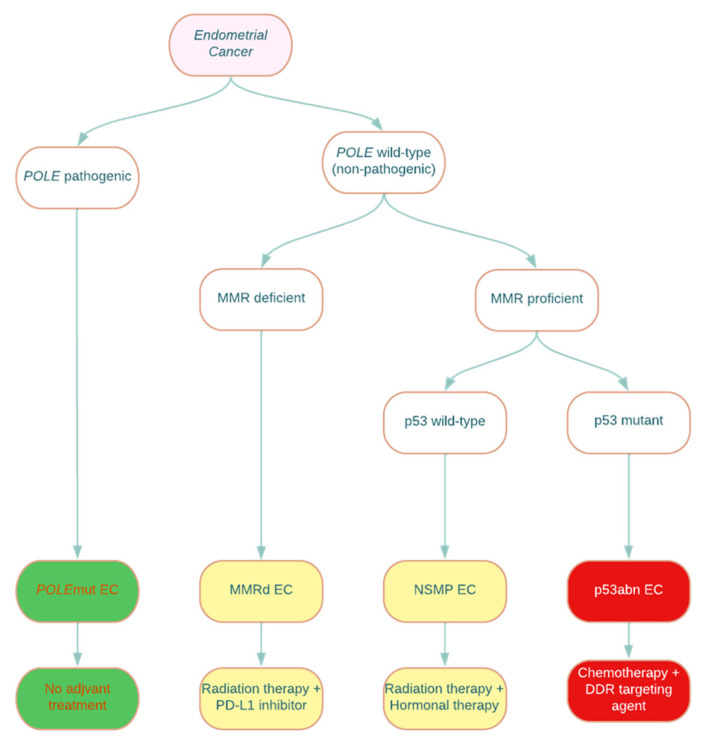
Molecular classification of endometrial cancer with associated suggested management. *POLE*: polymerase epsilon catalytic subunit; MMR: mismatch repair; *POLE*mut: polymerase epsilon-ultramutated; MMRd: mismatch repair deficient; NSMP: no specific molecular profile; p53abn: p53-abnormal; PD-L1: programmed death ligand-1; DDR: DNA damage response. The green subtype is associated with an excellent prognosis, the yellow subtype is associated with an intermediate prognosis, and the red subtype is associated with a poor prognosis [12].

**Figure 3 cancers-16-01869-f003:**
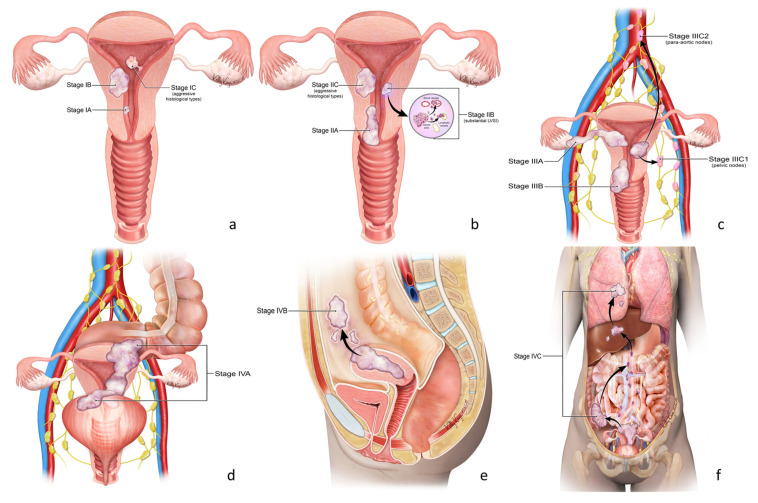
2023 FIGO staging of endometrium cancer (Table 1). Stage I (**a**). Confined to the uterine corpus and ovary. Stage II (**b**). Invasion of cervical stroma with extrauterine extension OR with substantial LVSI OR aggressive histological types with myometrial invasion. Stage III (**c**). Local and/or regional spread of the tumor of any histological subtype. Stage IV (**d**–**f**). Spread to the bladder and/or intestinal mucosa and/or distance metastasis.

**Figure 4 cancers-16-01869-f004:**
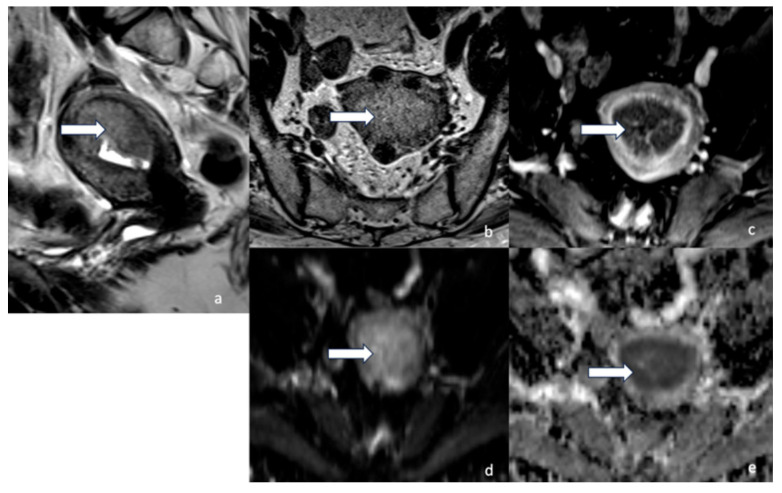
Stage IA2. A 53-year-old female patient with low-grade endometrioid carcinoma, sagittal T2 (**a**), axial T2 (**b**), axial post-contrast T1 (**c**), (**d**) DWI and (**e**) ADC MRI images show an ill-defined hypo-enhancing mass in the endometrial cavity (arrow) with restricted diffusion—the mass projects into less than 50% of the myometrium. Low-grade endometrioid carcinoma is a non-aggressive histological type, and when it involves less than half of myometrium, it is considered stage IA2.

**Figure 5 cancers-16-01869-f005:**
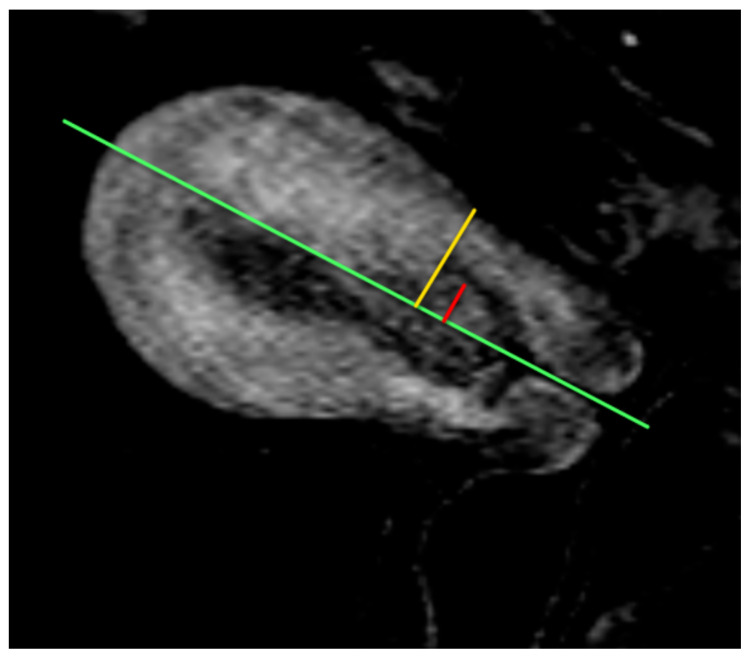
Sagittal post-contrast T1WI of the deepest extent of myometrial invasion. Measure the depth of myometrial invasion by drawing a line along the expected inner edge of the myometrium (green line). Then, draw lines measuring the thickness of the entire myometrium (yellow line) and the extent of tumor invasion into the myometrium (red line). The ratio of the red line over the yellow line is the percentage of myometrial invasion.

**Figure 6 cancers-16-01869-f006:**
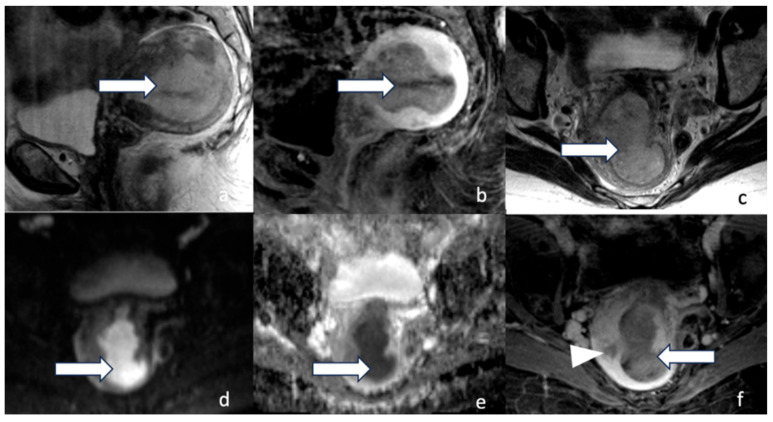
Stage IB. A 58-year-old female with low-grade endometrioid carcinoma. Sagittal T2 (**a**), sagittal post-contrast T1 (**b**), axial T2 (**c**), DWI (**d**), ADC (**e**), and axial post-contrast T1 (**f**), MRI images revealed a 5.3 cm hypo-enhancing mass (arrow) within the endometrium involving greater than 50% of the myometrium (arrowhead). Low-grade endometrioid carcinoma is a non-aggressive histological type. When it involves half or more than half of the myometrium and with no lymphovascular involvement, it is staged as stage IB.

**Figure 7 cancers-16-01869-f007:**
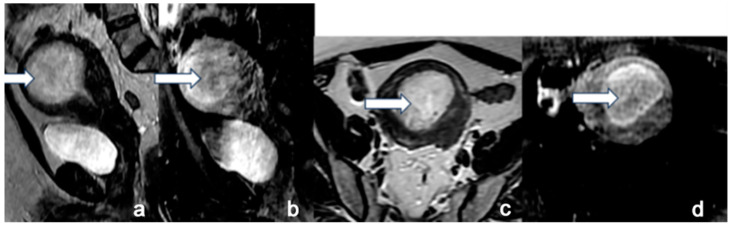
Stage IC. A 45-year-old female with uterine serous carcinoma. Sagittal T2 (**a**), sagittal post-contrast T1 (**b**), axial T2 (**c**), and axial post-contrast T1 (**d**) MRI images show a large expansile endometrial mass (white arrow) without myometrial invasion. The findings were confirmed on post-hysterectomy surgical pathology. Uterine serous carcinoma is an aggressive histological type, and when it is confined to the endometrium, it is staged as stage IC.

**Figure 8 cancers-16-01869-f008:**
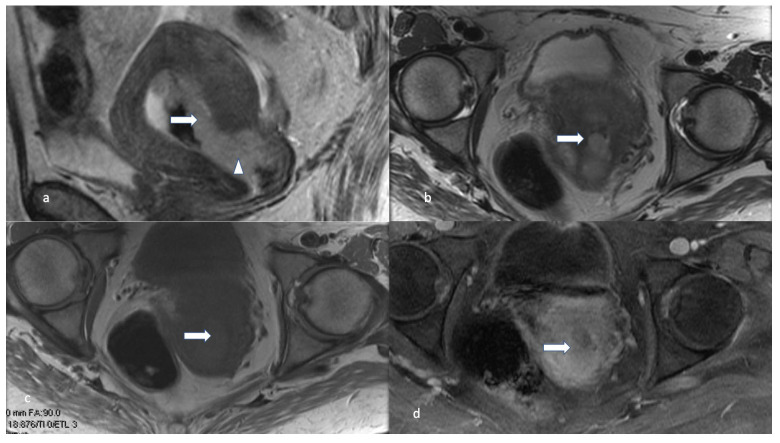
Stage IIA. A 55-year-old female with low-grade endometrioid carcinoma. Sagittal T2 (**a**), axial T2 (**b**), T1 pre (**c**), and post-contrast T1 (**d**) MRI images show a large endometrial mass (white arrow) measuring 3.0 × 1.8 cm and involves less than 50% of the myometrium. The mass extends into the endocervical canal (arrowhead) and involves the cervix. Invasion of the cervical stroma of the non-aggressive histological subtype is staged as stage IIA.

**Figure 9 cancers-16-01869-f009:**
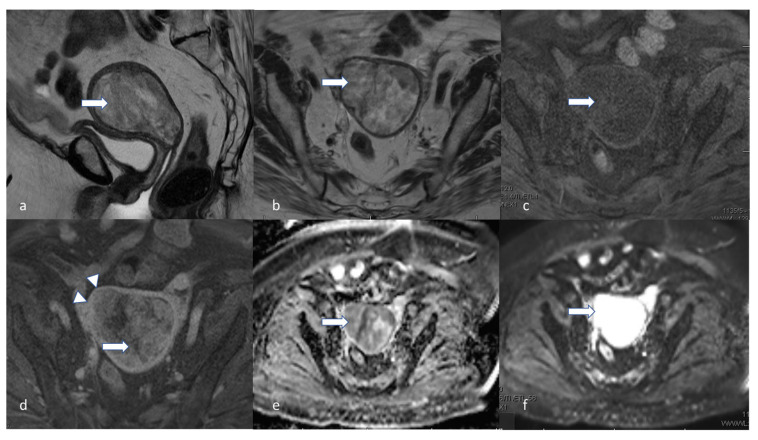
Stage IIC. A 60-year-old female patient with endometrial carcinosarcoma. Sagittal T2 (**a**), axial T2 (**b**), axial T1 pre (**c**), and post (**d**) contrast show a large heterogeneously enhancing mass (arrow) distending the entire endometrial cavity. The mass extends into the superior aspect of the endocervical canal. The mass appears to have a low ADC value (**e**) and a high signal in DWI (**f**). The mass invades the myometrial wall along the right side of the uterine fundus and along the right lateral wall of the uterine body (arrowheads). Endometrial carcinosarcoma is an aggressive histological type, and, hence, any myometrial involvement is staged as stage IIC.

**Figure 10 cancers-16-01869-f010:**
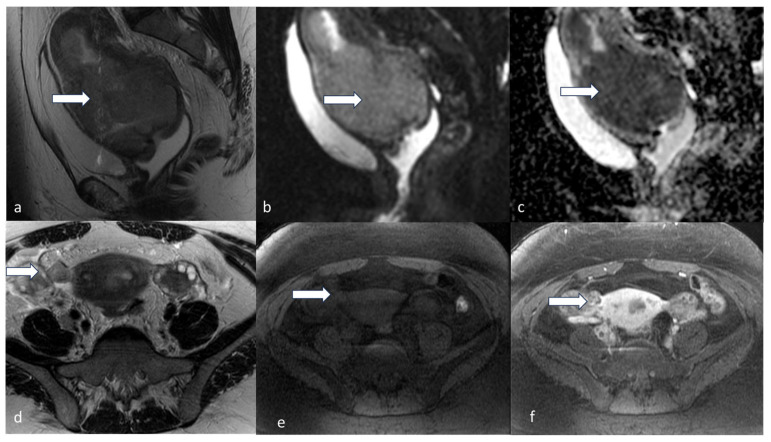
Stage IIIA1. A 42-year-old female patient with high-grade endometroid carcinoma. Sagittal T2 (**a**), sagittal DWI (**b**), and ADC (**c**) show a large heterogeneous uterine mass (arrow) involving the entire uterus with restricted diffusion. Axial T2 (**d**), axial T1 pre (**e**), and post-contrast TI (**f**), MRI images show a tumor deposit (arrow) in the region of the right fallopian tube/broad ligament, showing transtubal spread of the tumor to the right ovary.

**Figure 11 cancers-16-01869-f011:**
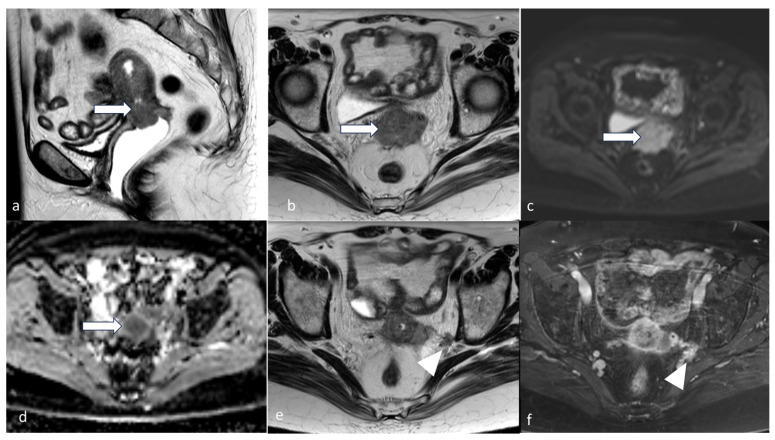
Stage IIIB2_POLEmut_. A 61-year-old female with post-menopausal bleeding with high-grade endometrioid carcinoma. Sagittal T2 (**a**) and axial T2 (**b**) MRI images show diffuse thickening with mass-like nodularity (arrow) in the lower uterine segment measuring. The mass shows a high DWI (**c**) signal and low ADC value (**d**), denoting diffusion restriction. Axial T2 (**e**) and axial T1 fat sat post-contrast (**f**) show a tumor deposit along the left pelvic sidewall abutting the left obturator internus muscle (arrowhead). The molecular subtype was reported to be polymerase epsilon-ultramutated (POLEmut).

**Figure 12 cancers-16-01869-f012:**
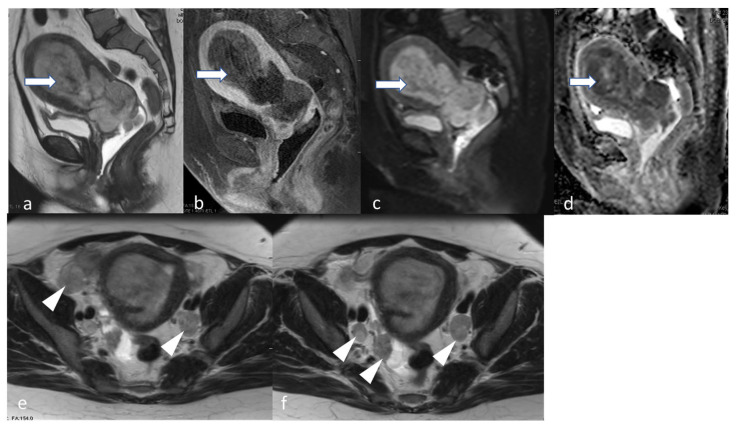
Stage IIIC1ii. A 48-year-old female patient with high-grade endometrial carcinoma. Sagittal T2 (**a**) and T1 fat sat post-contrast (**b**) show a large mildly T2 hyperintense hypovascular endometrial mass (arrow) causing expansion of the endometrial cavity with diffusion restriction evidenced by a relatively high signal in DWI (**c**) and low ADC value (**d**). The mass extends inferiorly through the cervical canal involving the posterior lip and the posterior final fornix. Axial T2 images (**e**,**f**) show bilateral iliac metastatic lymph nodes (arrowhead).

**Figure 13 cancers-16-01869-f013:**
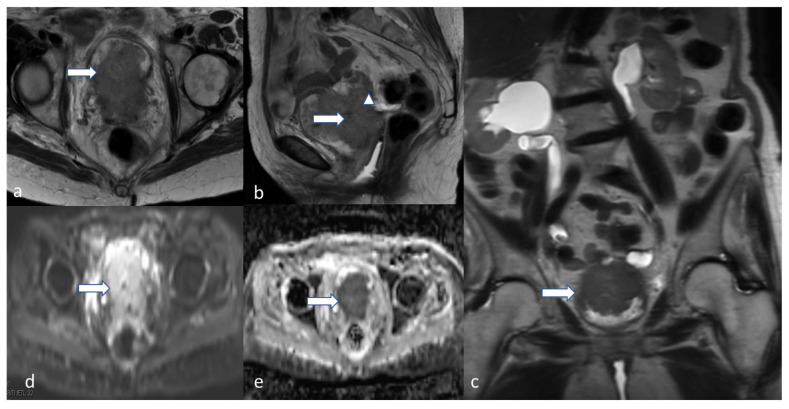
Stage IVA. A 91-year-old female with clear cell endometrial carcinoma. MRI axial T2 (**a**) and sagittal T2 (**b**) weighted MRI images show a large endometrial mass (arrow). This extends to the vaginal cuff, the rectosigmoid (arrowhead in image (**b**)), anteriorly invades through the posterior urinary bladder wall. The mass involves bilateral ureterovesical junctions and distal ureters, causing bilateral hydronephrosis (coronal T2, image (**c**)). The mass exhibits diffusion restriction with a mildly bright signal in DWI (**d**) and a low signal in ADC map (**e**). This patient’s FIGO 2009 vs. FIGO 2023 staging is the same.

**Figure 14 cancers-16-01869-f014:**
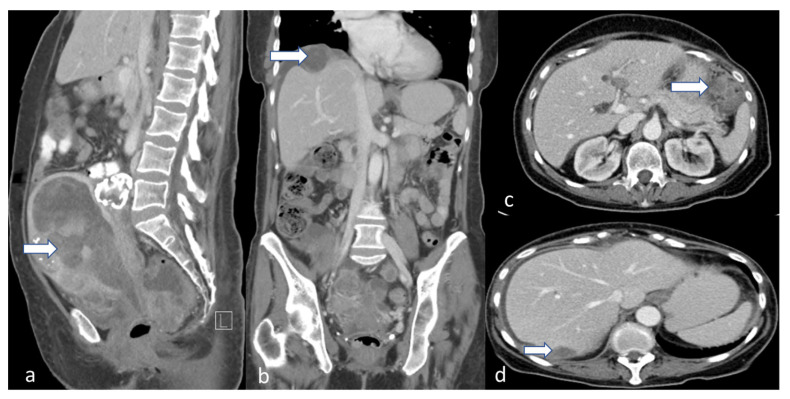
Stage IVB. A 65-year-old female with grade 2 low-grade endometrioid carcinoma. Sagittal (**a**), coronal (**b**), and axial (**c**,**d**) post-contrast CT venous phase images show an enlarged uterus with a large endometrial mass (arrow). Peritoneal implants are noted along the right subdiaphragmatic/perihepatic implant space and in the left upper quadrant (arrow in (**b**–**d**)).

**Figure 15 cancers-16-01869-f015:**
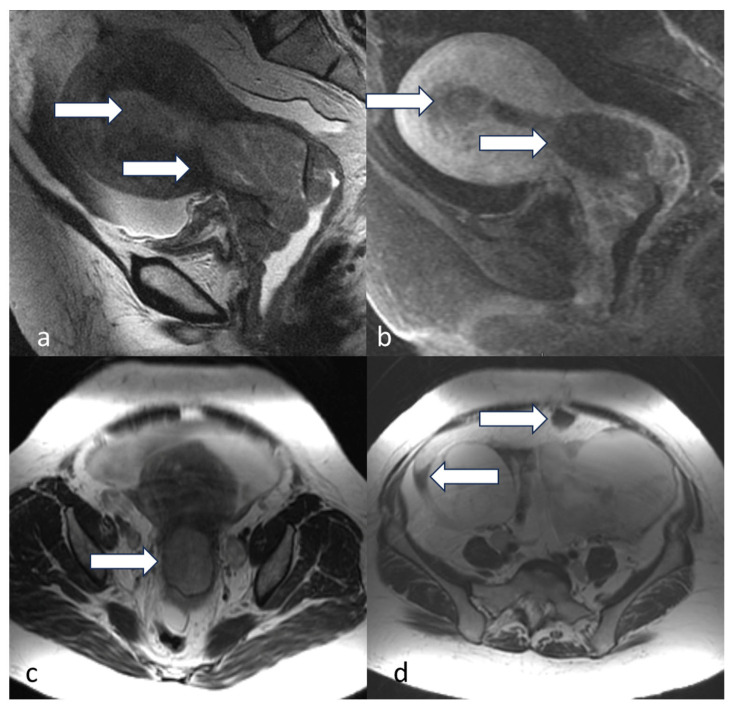
Stage IVB. A 52-year-old female with serous endometrial carcinoma. Sagittal T2 (**a**), sagittal T1 post-contrast (**b**), and axial T2 (**c**) weighted MRI images show thickened irregular endometrium (arrow) with involvement of the cervix as well as the upper vagina. Axial (**d**) T2 weighted MRI image peritoneal involvement with moderate-volume ascites and peritoneal implant (arrow).

**Figure 16 cancers-16-01869-f016:**
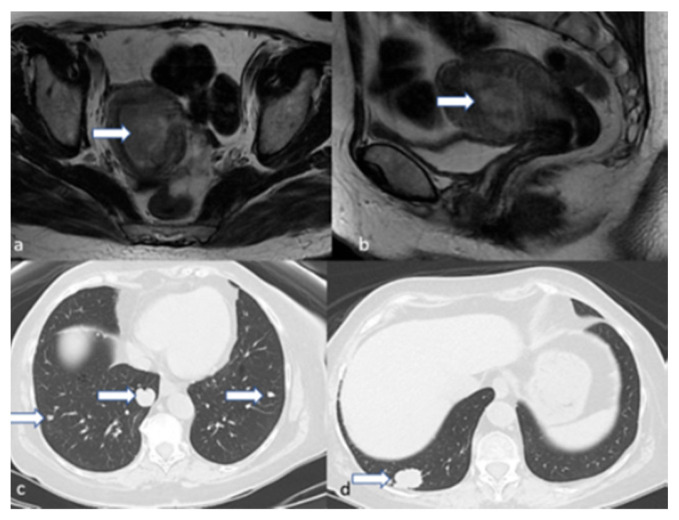
Stage IVC. A 44-year-old female patient with high-grade endometrioid carcinoma. Axial (**a**) and sagittal (**b**) T2 weighted MRI images show an endometrial mass (arrow). Axial CT of the chest (**c**,**d**) revealed multiple pulmonary metastatic nodules (arrows); the largest is at the right lower lobe.

**Figure 17 cancers-16-01869-f017:**
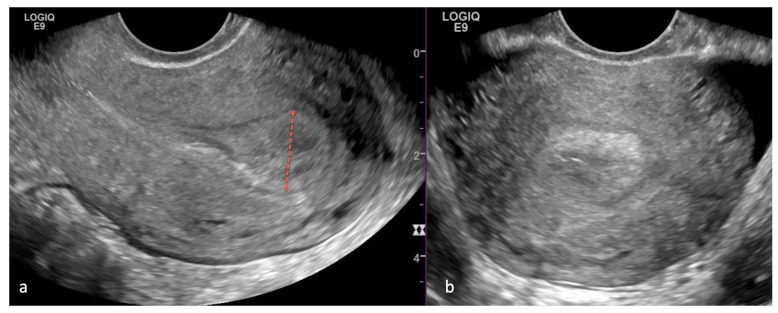
A 52-year-old woman with postmenopausal bleeding. Sagittal (**a**) and axial (**b**) transvaginal ultrasound images of the uterus demonstrate a thickened endometrium measuring 15 mm (red dotted line). Endometrial biopsy was performed with the diagnosis of high-grade endometroid carcinoma cancer.

**Figure 18 cancers-16-01869-f018:**
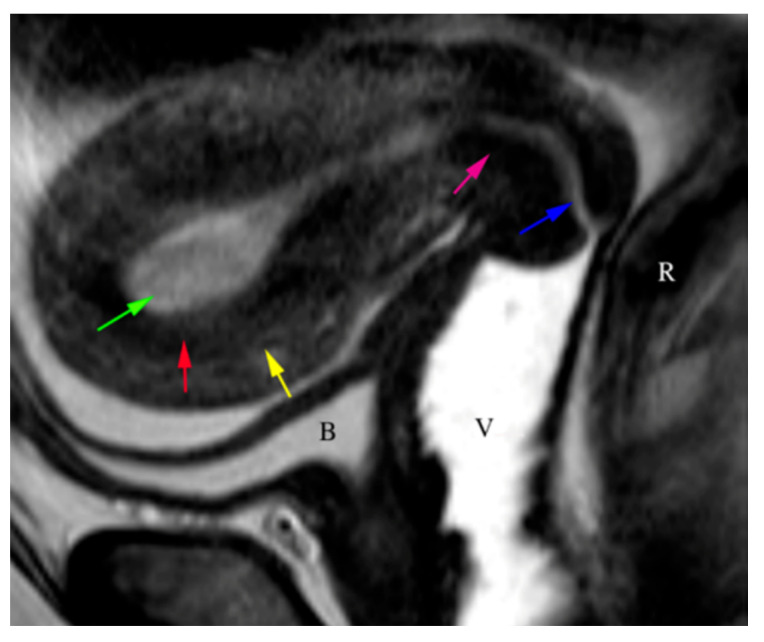
The sagittal T2WI image demonstrates the normal trilaminar appearance of the uterus. The T2 hyperintense endometrium (green arrow), the hypointense junctional zone (red arrow), and the isointense myometrium (yellow arrow) create the trilaminar appearance. The cervix demonstrates a hyperintense endocervical canal (blue arrow) and hypointense cervical (pink arrow) stroma. Bladder (B), vagina (V), rectum (R).

**Figure 19 cancers-16-01869-f019:**
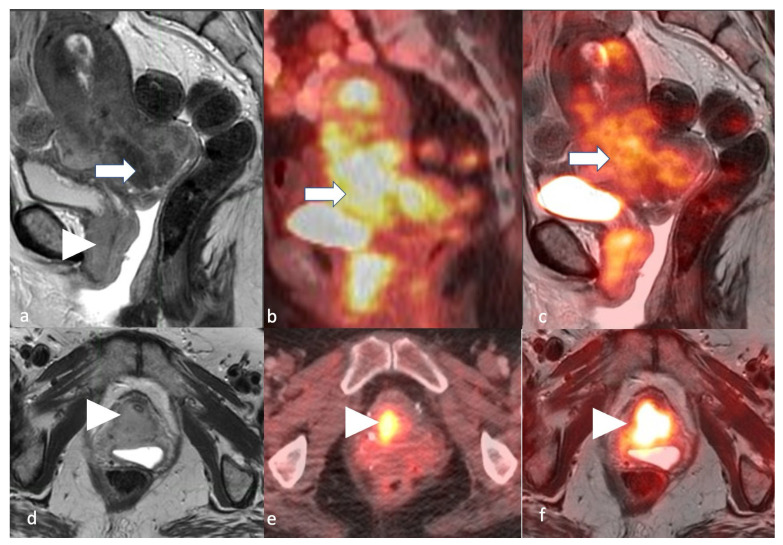
Stage IV. A 54-year-old female patient with endometrial carcinoma. Sagittal (**a**) and axial (**d**) T2 MRI images, sagittal (**b**) and axial (**e**) PET/CT images, and sagittal (**c**) and axial (**f**) T2-weighted PET/MRI images demonstrate an FDG avid (arrow) T2 intermediate hyperintense endometrial tumor involving the bladder base and urethra (arrowhead).

**Figure 20 cancers-16-01869-f020:**
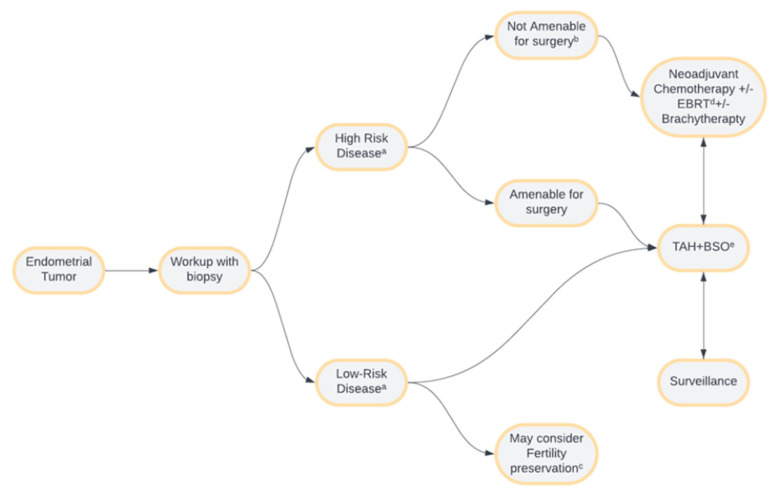
Endometrial cancer management overview. ^a^ High-risk disease involves any stage with an aggressive subtype and/or stage III or IV disease regardless of histological subtype. Low-risk disease involves stage I or stage II disease with a non-aggressive subtype. ^b^ Disease deemed unresectable by surgical evaluation or late-stage disease where risks of surgery outweigh the benefits. ^c^ Fertility preservation, though controversial, may only be considered in early-stage, low-risk diseases. ^d^ External beam radiation therapy. ^e^ Total abdominal hysterectomy with bilateral salpingo-oophorectomy.

**Table 1 cancers-16-01869-t001:** 2023 FIGO staging of endometrial cancer [4].

Stages	Description
Stage I	Confined to uterine corpus and ovary ^a^
IA	Disease limited to the endometrium OR non-aggressive histological type, with < 50% myometrial involvement, no focal LVSI ^b^, or good disease prognosis.
IA1	Non-aggressive histological type limited to an endometrial polyp or confined to endometrium.
IA2	Non-aggressive histological type with < 50% myometrial involvement with no to focal LVSI ^b^.
IA3	Low-grade endometrioid carcinomas limited to uterus or ovary ^a^.
IB	Non-aggressive histological type with > 50% myometrial invasion, and no to focal LVSI ^b^.
IC	Aggressive histological subtype limited to a polyp or confined to endometrium.
Stage II	Invasion of cervical stroma with no extrauterine extension, or substantial LVSI ^b^, or aggressive histological type with any myometrial invasion.
IIA	Non-aggressive histological type with invasion of cervical stroma.
IIB	Non-aggressive histological type with substantial LVSI ^b^.
IIC	Aggressive histological subtype with any myometrial involvement.
Stage III	Local and/or regional metastasis of any histological subtype.
IIIA	Invasion of uterine serosa, adnexa, or both.
IIIA1	Spread to ovary or fallopian tube (excluding lesions that meet stage IA3 criteria ^a^).
IIIA2	Spread to uterine subserosa or through uterine serosa.
IIIB	Metastasis or direct spread to the vaginal canal and/or parametria or pelvic peritoneum.
IIIB1	Metastasis or direct spread to the vaginal canal and/or parametria.
IIIB2	Metastasis to pelvic peritoneum.
IIIC	Metastasis to pelvic and/or para-aortic lymph nodes.
IIIC1	Metastasis to pelvic lymph nodes.
IIIC1i	Micrometastasis to pelvic lymph nodes.
IIIC1ii	Macrometastasis to pelvic lymph nodes.
IIIC2	Metastasis to para-aortic lymph nodes up to the renal vessels, and/or pelvic lymph node metastasis.
IIIC2i	Micrometastasis to para-aortic lymph nodes up to the renal vessels, and/or pelvic lymph node metastasis.
IIIC2ii	Macrometastasis to para-aortic lymph nodes up to the renal vessels, and/or pelvic lymph node metastasis.
Stage IV	Spread to the bladder mucosa and/or intestinal mucosa and/or distant metastasis.
IVA	Spread to the bladder mucosa and/or intestinal mucosa.
IVB	Abdominal peritoneal metastasis beyond the pelvic peritoneum,
IVC	Distant metastasis, included metastasis to any extra-abdominal or intra-abdominal lymph nodes superior to the renal vessels, lungs, liver, brain, or bone.

^a^ Concurrent uterine and ovarian low-grade endometrioid endometrial carcinoma are included in this stage if the following criteria are met: (1) No more than 50% of myometrial involvement, (2) absence of extensive LVSI, (3) absence of additional metastasis, and (4) unilateral ovarian involvement without capsule rupture or invasion. ^b^ Lymphovascular invasion.

**Table 2 cancers-16-01869-t002:** Comparison of FIGO staging for stage I.

	2009	2023
Stage IA	<50% invasion of the myometrium or no invasion	Limited to endometrium with no or focal LVSI IA1 Non-aggressive, limited to polyp or endometrium IA2 Non-aggressive, involving less than ½ of myometrium with or without focal LVSI IA3 Low-grade endometroid carcinoma limited to the uterus and ovary
Stage IB	Invasion of >50% of the myometrium	IB Non-aggressive, with invasion of more than ½ of the myometrium with or without focal LVSI
Stage IC		IC Aggressive type limited to a polyp or confined to the endometrium

**Table 3 cancers-16-01869-t003:** Comparison of FIGO staging for stage II.

	2009	2023
Stage II	Tumor invading the cervical stroma but within the uterus	Invasion of cervix with extrauterine extension and extensive LVSI OR aggressive type with myometrial invasion IIA Non-aggressive type with invasion of cervix stroma IIB Substantial LVSI of non-aggressive type IIC Aggressive type with any myometrial involvement

**Table 4 cancers-16-01869-t004:** Comparison of FIGO staging for stage III.

	2009	2023
Stage III	Tumor spread locally or regionally, extrauterine disease present	Aggressive or non-aggressive extrauterine disease
Stage IIIA	Invasion of adnexa or uterine serosa	IIIA1 Spread to ovary or fallopian tube IIIA2 Spread to uterine subserosa or through uterine serosa
Stage IIIB	Invasion of parametrium and/or vagina	Metastasis to vagina and/or parametria or pelvic peritoneum IIIB1 Direct spread to vagina or parametria IIIB2 Metastasis to pelvic peritoneum
Stage IIIC	Invasion of pelvic and/or para-aortic lymph nodes IIIC1 Metastasis to pelvic lymph nodes	Metastasis to pelvic or para-aortic lymph nodes or both IIIC1 Metastasis to pelvic lymph nodes IIIC1i Micrometastasis IIIC1ii Macrometastasis
	IIIC2 Metastasis to para-aortic lymph nodes w/ or w/o spread to pelvic lymph nodes	IIIC2 Metastasis to para-aortic lymph nodes w/ or w/o spread to pelvic lymph nodes IIIC2i Micrometastasis IIIC2ii Macrometastasis

**Table 5 cancers-16-01869-t005:** Comparison of FIGO staging for stage IV.

	2009	2023
Stage IV	Rectal or bladder involvement and/or distant metastasis present	Invasion of cervix with extrauterine extension and extensive LVSI OR aggressive type with myometrial invasion
Stage IVA	IVA Invasion of bladder or rectum	IVA Invasion of bladder and/or bowel mucosa
Stage IVB	IVB Invasion of other organs and/or lymph nodes	IVB Abdominal peritoneal metastasis
Stage IVC		IVC Metastasis to extra- or intra-abdominal lymph nodes or organs

**Table 6 cancers-16-01869-t006:** Pelvic MRI Protocol.

Series Description	Slice Thickness (mm)	Inter-Slice Gap (mm)	FOV	Acquisition Time (min)	Frequency Encoding Direction	Acquired Matrix (Frequency × Phase)	*b*-Value (s/mm^2^)	Coverage
Coronal T2 weighted single-shot FSE (to include the kidneys)	5	0	420	1.4	Superior–inferior	288 × 192		Entire pelvis to include the kidneys
Sagittal T2 weighted (FSE)	5	0	240	2	Anterior–posterior	320 × 224		Entire uterus, cervix
Sagittal restricted—FOV DW images with spin echo	5	0	240	16	Superior–inferior	96 × 80	50, 600	Entire uterus, cervix
Axial T2 weighted FSE	5	0	240	2	Left–right	320 × 224		Entire pelvis
Axial T1 weighted FSE	5	0	240	2	Left–right	320 × 224		Entire pelvis
Axial DW images with spin echo	5	0	380	8	Left–right	96 × 160	50, 400, 800	Entire pelvis
Axial 3D unenhanced T1-weighted spoiled gradient-recalled echo with fat suppression	5	−2.5	240	1	Left–right	320 × 224		Entire pelvis
Sagittal T1-weighted dynamic contrast-enhanced spoiled gradient-recalled echo with fat suppression	5	−2.5	240	1	Superior–inferior	256 × 224		Entire uterus, cervix
Three-dimensional contrast-enhanced T1-weighted spoiled gradient-recalled echo with fat suppression	5	−2.5	240	1	Left–right	320 × 224		Entire pelvis

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
