# Peer review of "Endometrial Cancer: 2023 Revised FIGO Staging System and the Role of Imaging"

_cancers, 2024, doi:10.3390/cancers16101869_

Round 1
Reviewer 1 Report
Comments and Suggestions for Authors
The submission Endometrial Cancer: 2023 Revised FIGO Staging System and the Role of Imaging." starts with a good premise that imaging considerations should be incorporated in the FIGO standards as they have value.
First, the readership who may best benefit from this work are radiologists, clinical scientists, etc., bringing imaging to the discussion's forefront. Or perhaps it is the surgeons, and oncologists who would benefit by understanding the impact of imaging and if that is the case, the authors should provide stronger reasoning on this issue. For example, do gynecology surgeons and oncologists rely on imaging (more than ultrasound) at your or other institutions if so, the authors should carefully explain why and/or why not.
The sentence on the line ". Although not included in the staging classification, imaging studies are crucial for the accurate screening, diagnosis, staging, and follow-up of Endometrial Cancer (EC). Different imaging modalities are beneficial across various stages of EC management." This is a complex sentence and unclear. The authors should rewrite this sentence into two sentences ... as the sentence is ambiguous as constructed. (line 22,23)
2) The authors then switch and bring in a discussion regarding histopathology which detracts from the line of thinking on imaging in the paper which according to the title of this article is the focus
The paper should be rewritten to clarify how the identification of molecular subtypes and imaging correlate as relationships between these domains (the authors should describe those correlates if these exist or state that it’s an active area for investigation). For example, one suggestion is that perhaps there is imaging that is a marker of aggressiveness that pathologist can use in their interpretations and/or vice versa. This could be explored in greater detail, otherwise, organizationally this is out of place.
3) Figure 3 is a good drawing of the stages of Endometrial Cancer and the anatomical spaces it involves. However, did the authors create this illustration, or is it from a stock source? Attribution should be mentioned.
The authors should bring out the LVSI and myometrial invasion components in these illustrations as it is the key point that these are two major branches for imaging review. The authors should then also provide a more detailed discussion on implications and understanding of imaging to create a more comprehensive picture of the disease from an imaging/radiological point of view.
4) The reviewer suggests that the author review resources such as Update on MRI in Evaluation and Treatment of Endometrial Cancer (rsna.org), Update on MRI in Evaluation and Treatment of Endometrial Cancer | RadioGraphics (rsna.org)and even Endometrial carcinoma (staging) | Radiology Reference Article | Radiopaedia.org, Role of MRI in staging and follow-up of endometrial and cervical cancer: pitfalls and mimickers | Insights into Imaging | Full Text (springeropen.com) and Endometrial carcinoma: Clinical features, diagnosis, prognosis, and screening - UpToDate, etc. The references and combined knowledge and information could be more thoroughly reviewed.
The reason this is mentioned is that the article should allow the authors to make a strong case for inclusion of imaging into future guidelines and strong advocacy would dictate a solid image review.
5) Table 1 is a reprinting of FIGO standards for 2023, is there a permissions issue? Some statements and discussion on the value of reprinting this information should be provided in the article.
6) section 4.1 appears to begin the discussion of imaging in this work (which may be more important to start earlier in this paper – as mentioned above they jump into histopathology without coherence on imaging) and this is where the reviewer could bring some of the themes and ideas to the forefront. The goal for radiology is perhaps for clinical staging (TMN), disease description (so referring can use this), treatment evaluation(to inform treatment pathways whether resection and/or systemic), and prognosis determination (staging, etc. ). So organization is the key, and all such discussions should be brought out clearly in this work.
7) Much more effort explaining the images and background and why they are good examples needs to be done to expand upon the value for staging from lines 130 to 280. It is effortful to read these sections and it is not clearly 'teaching' readership the value of imaging in these cases. Please consider comparing your images to other publications in this area and what is more demonstrative and useful for understanding its importance.
8) Table 6 is useful as a resource, but needs more parameters especially the number of slices, positioning, and time of the scan. If this can’t fit in the paper as is, this should go in the supplemental information. How often MRIs get ordered for endometrial cancer at the author’s institution should be mentioned.
Comments on the Quality of English LanguageThis publication requires some editorial work; it would be beneficial to engage a proofreader to enhance clarity. If authors have access to such services, we as authors should utilize them to help us.
Author Response
|
Response to Reviewer 1 Comments
|
||
|
1. Summary |
|
|
|
Thank you very much for taking the time to review this manuscript. Please find the detailed responses below and the corresponding revisions/corrections highlighted/in track changes in the re-submitted files.
|
||
|
2. Questions for General Evaluation |
|
|
|
3. Point-by-point response to Comments and Suggestions for Authors |
|
|
|
Comments 1: First, the readership who may best benefit from this work are radiologists, clinical scientists, etc., bringing imaging to the discussion's forefront. Or perhaps it is the surgeons, and oncologists who would benefit by understanding the impact of imaging and if that is the case, the authors should provide stronger reasoning on this issue. For example, do gynecology surgeons and oncologists rely on imaging (more than ultrasound) at your or other institutions if so, the authors should carefully explain why and/or why not.
The sentence on the line ". Although not included in the staging classification, imaging studies are crucial for the accurate screening, diagnosis, staging, and follow-up of Endometrial Cancer (EC). Different imaging modalities are beneficial across various stages of EC management." This is a complex sentence and unclear. The authors should rewrite this sentence into two sentences ... as the sentence is ambiguous as constructed. (line 22,23)
|
||
|
Response 1: Thank you for the comment. The target audience of this article is multidisciplinary, as is the diagnosis and treatment of gynecological cancers. Because of this, the authors discuss the important changes in FIGO 2023 to then present the importance of radiology. Currently at our institution, tumor board meetings are crucial for discussing topics such as the evolving diagnosis and management of endometrial cancer. The radiologists’ role in these meetings is equally paramount to the gynecological surgeons and oncologists in guiding best management and diagnostic practices.
The authors agree that this sentence may cause confusion as currently written. This sentence was rephrased to address the other point in this comment to clarify the audience and now reads as follows: “Gynecological cancer is a crucial element in the practice of a body radiologist. With a new diagnostic system in place, it is important to address the role of radiology in EC diagnostic pathway.” [Lines 26-28]
|
||
|
Comments 2: The authors then switch and bring in a discussion regarding histopathology which detracts from the line of thinking on imaging in the paper which according to the title of this article is the focus The paper should be rewritten to clarify how the identification of molecular subtypes and imaging correlate as relationships between these domains (the authors should describe those correlates if these exist or state that it’s an active area for investigation). For example, one suggestion is that perhaps there is imaging that is a marker of aggressiveness that pathologist can use in their interpretations and/or vice versa. This could be explored in greater detail, otherwise, organizationally this is out of place.
|
||
|
Response 2: Thank you for the thorough feedback on our work. The direct histopathological/ molecular correlation of imaging findings in cancer is an active field in research. We agree that this approach to gynecological cancers such as endometrial cancer is an unexplored opportunity in research. Due to the current lack of evidence, the authors have made the best approach a multidisciplinary discussion of the current FIGO guidelines, highlighting the importance and pitfalls of radiology’s role.
Comments 3: Figure 3 is a good drawing of the stages of Endometrial Cancer and the anatomical spaces it involves. However, did the authors create this illustration, or is it from a stock source? Attribution should be mentioned.
Response 3: Thank you for the comment, this illustration was created exclusively for the use in this article. The medical illustrator, Kelly Page, was included in the acknowledgements section of the article.
Comments 4: The reviewer suggests that the author review resources such as Update on MRI in Evaluation and Treatment of Endometrial Cancer (rsna.org), Update on MRI in Evaluation and Treatment of Endometrial Cancer | RadioGraphics (rsna.org) and even Endometrial carcinoma (staging) | Radiology Reference Article | Radiopaedia.org, Role of MRI in staging and follow-up of endometrial and cervical cancer: pitfalls and mimickers | Insights into Imaging | Full Text (springeropen.com) and Endometrial carcinoma: Clinical features, diagnosis, prognosis, and screening - UpToDate, etc. The references and combined knowledge and information could be more thoroughly reviewed. The reason this is mentioned is that the article should allow the authors to make a strong case for inclusion of imaging into future guidelines and strong advocacy would dictate a solid image review.
Response 4: Thank you for providing these additional resources to consider. We the authors agree that these references are invaluable to our topic discussion. As such, we include either the direct reference listed, or the sources material that was used to create these resources.
Comments 5: Table 1 is a reprinting of FIGO standards for 2023, is there a permissions issue? Some statements and discussion on the value of reprinting this information should be provided in the article.
Response 5: Table 1 has been created by the author and adapted and reference the published FIGO 2023 guidelines for Endometrial Cancer.
Comments 6: Section 4.1 appears to begin the discussion of imaging in this work (which may be more important to start earlier in this paper – as mentioned above they jump into histopathology without coherence on imaging) and this is where the reviewer could bring some of the themes and ideas to the forefront. The goal for radiology is perhaps for clinical staging (TMN), disease description (so referring can use this), treatment evaluation (to inform treatment pathways whether resection and/or systemic), and prognosis determination (staging, etc.). So, organization is the key, and all such discussions should be brought out clearly in this work.
Response 6: Section 4.1 begins the discussion of the available evidence of the role of each imaging modality in the diagnosis of endometrial cancer. The prior sections highlight the changes to the classification system, discussing the respective relevance of imaging studies per stage.
Comments 7: Much more effort explaining the images and background and why they are good examples needs to be done to expand upon the value for staging from lines 130 to 280. It is effortful to read these sections and it is not clearly 'teaching' readership the value of imaging in these cases. Please consider comparing your images to other publications in this area and what is more demonstrative and useful for understanding its importance.
Response 7: Thank you for your feedback. The authors have taken careful consideration in the selection and curation of the multimodality imaging examples. We hope that the selected images highlight key findings in each endometrial cancer stage that are crucial for radiologists to acknowledge.
Comments 8: Table 6 is useful as a resource, but needs more parameters especially the number of slices, positioning, and time of the scan. If this can’t fit in the paper as is, this should go in the supplemental information. How often MRIs get ordered for endometrial cancer at the author’s institution should be mentioned.
Response 8: The authors agree that these are important parameters to include in our review. An additional column was added to table 6, displaying the acquisition time per sequence at our institution. Regarding the position, in section 5.3 we specify that that most beneficial technique is an axial oblique MRI angled to the endometrial cavity. This position allows for a better assessment of myometrial invasion. There is no standard slice amount in our institution, but we emphasize the importance of slice thickness and gap, to achieve an appropriate balance of scanned tissue coverage and minimal signal overlap.
|
||
|
4. Response to Comments on the Quality of English Language |
||
|
Point 1: This publication requires some editorial work; it would be beneficial to engage a proofreader to enhance clarity. If authors have access to such services, we as authors should utilize them to help us. |
||
|
Response 1: Thank you for the recommendations. We have used resources available in our institution to improve the language quality in our manuscript, guided by the feedback provided by the reviewers. |
||
Reviewer 2 Report
Comments and Suggestions for Authors
This review described roles of radiological imaging corresponding to the revised 2023 FIGO staging of endometrial cancer. The revised 2023 FIGO staging provides more accurate definitions of prognostic groups to direct appropriate management, aiming to enhance patient care significantly. The revised staging emphasizes aggressive and non-aggressive histology, and more detailed lymphovascular invasion status in the staging of endometrial cancer. This review delineated 2009 and 2023 FIGO staging differences in detail by comparing corresponding stages and sub-stages.
MRI imaging examples were demonstrated for stage I, II, III and IV of endometrial cancer. Imaging modalities of ultrasound, CT, MRI, PET/CT and PET/MRI were discussed. Their usages, indications, sensitivity, and specificities in cancer staging and evaluation of advanced diseases were described surely.
The review also provided brief insights of molecular classification of endometrial cancer based on the Cancer Genome Atlas. The molecular classification together with imaging and surgical staging contribute to more advanced management of the cancer.
This article provides a comprehensive review of changes in the revised 2023 FIGO staging of endometrial cancer, focusing on imaging studies in cancer screening, diagnosis, staging and follow-up.
Please indicate geographic range of “By the end of 2023, there will be an estimated 66,200 new cases and 13,030 related deaths” (line 35).
Author Response
|
Comments 1: This review described roles of radiological imaging corresponding to the revised 2023 FIGO staging of endometrial cancer. The revised 2023 FIGO staging provides more accurate definitions of prognostic groups to direct appropriate management, aiming to enhance patient care significantly. The revised staging emphasizes aggressive and non-aggressive histology, and more detailed lymphovascular invasion status in the staging of endometrial cancer. This review delineated 2009 and 2023 FIGO staging differences in detail by comparing corresponding stages and sub-stages. MRI imaging examples were demonstrated for stage I, II, III and IV of endometrial cancer. Imaging modalities of ultrasound, CT, MRI, PET/CT and PET/MRI were discussed. Their usages, indications, sensitivity, and specificities in cancer staging and evaluation of advanced diseases were described surely. The review also provided brief insights of molecular classification of endometrial cancer based on the Cancer Genome Atlas. The molecular classification together with imaging and surgical staging contribute to more advanced management of the cancer. This article provides a comprehensive review of changes in the revised 2023 FIGO staging of endometrial cancer, focusing on imaging studies in cancer screening, diagnosis, staging and follow-up. Please indicate geographic range of “By the end of 2023, there will be an estimated 66,200 new cases and 13,030 related deaths” (line 35).
|
|
Response 1: Thank you for the comments, the statistics mentioned in this section refer to projections in the United States. This clarification was included in the manuscript. |
Reviewer 3 Report
Comments and Suggestions for Authors
1. Is there any role of hysteroscopy and laparoscopy in the imaging / staging system of endometrium cancer?
2. How about the value of blood flow examination (including resistance index) in the ultrasonography evaluation of endometrium cancer?
3. How about the effect of concurrent chemo-radiotherapy in the adjuvant treatment for endometrium cancer?
4. There are typo errors in the following---
(1), at lines 168-169: and axial post-contrast T1“(e)” à (f)
(2), at line 430: stage “IIB” if >50% of myometrium invasion. à IB
(3), at line 247: posterior “final” fornix à vaginal
(4), at line 257, Table 5: “Invasion of cervix with extrauterine extension and extensive LVSI or Aggressive type with myometrial invasion”
à Spread to the bladder mucosa and/or intestinal mucosa and/or distant metastasis.
Author Response
|
Comments 1: Is there any role of hysteroscopy and laparoscopy in the imaging / staging system of endometrium cancer?
|
|
Response 1: Hysteroscopy and laparoscopy are very important procedures for diagnosis and management of Endometrial Cancer. In diagnosis, their main contribution is to gather a biopsy specimen to undergo histopathological and molecular analysis. Imaging has an integral role in the diagnostic pathways alongside these procedures.
Comments 2: How about the value of blood flow examination (including resistance index) in the ultrasonography evaluation of endometrium cancer?
Response 2: Thank you for the comment. Changes in blood flow are related to angiogenesis that occurs during the natural progression of Endometrial Cancer. There is ongoing research aimed towards studying doppler parameters such as restrictive index, pulse index and peak systolic velocity to establish the role transvaginal doppler ultrasound in predicting EC staging. The common limitations in these studies are the user dependence of and technological variation between study centers.
Comments 3: How about the effect of concurrent chemo-radiotherapy in the adjuvant treatment for endometrium cancer?
Response 3: Concurrent management strategies have certainly been proved to be beneficial in certain cases. The main focus of our manuscript was to discuss radiology’s role in the evolving endometrial cancer diagnostic guidelines. However, a shift in the diagnostic guidelines impact management and was the reason for including a brief explanation of endometrial cancer treatment.
Comments 4: There are typo errors in the following--- (1), at lines 168-169: and axial post-contrast T1“(e)” à (f) (2), at line 430: stage “IIB” if >50% of myometrium invasion. à IB (3), at line 247: posterior “final” fornix à vaginal (4), at line 257, Table 5: “Invasion of cervix with extrauterine extension and extensive LVSI or Aggressive type with myometrial invasion” à Spread to the bladder mucosa and/or intestinal mucosa and/or distant metastasis.
Response 4: Thank you for the comment, these errors were corrected as recommended in the manuscript. |
Reviewer 4 Report
Comments and Suggestions for Authors
I read the article by Menedez Santos et al. titled 'Endometrial Cancer: 2023 Revised FIGO staging system and the role of Imaging' with great interest.
The authors aimed to re-examine the FIGO 2023 staging of endometrial cancer from a radiological imaging perspective.
The new FIGO classification is a new classification: as a physician I know that it is not yet in routine use.
The classification is commendable as it incorporates aspects of molecular biology into a single classification, as many other articles have noted in the past.
In this paper, the authors have made a correlation between the new FIGO classification and radiological imaging. The article is well-written and fluent, with Table 6 being particularly interesting. It would be beneficial for non-radiologists to have a table with practical tips on how to read the images easily and critically, especially for radiotherapy oncologists like myself who frequently use radiological knowledge.
In conclusion, the work is of high quality, even in its current presentation.
Author Response
|
Comments 1: I read the article by Menendez Santos et al. titled 'Endometrial Cancer: 2023 Revised FIGO staging system and the role of Imaging' with great interest. The authors aimed to re-examine the FIGO 2023 staging of endometrial cancer from a radiological imaging perspective. The new FIGO classification is a new classification: as a physician I know that it is not yet in routine use. The classification is commendable as it incorporates aspects of molecular biology into a single classification, as many other articles have noted in the past. In this paper, the authors have made a correlation between the new FIGO classification and radiological imaging. The article is well-written and fluent, with Table 6 being particularly interesting. It would be beneficial for non-radiologists to have a table with practical tips on how to read the images easily and critically, especially for radiotherapy oncologists like myself who frequently use radiological knowledge. In conclusion, the work is of high quality, even in its current presentation.
|
|
Response 1: Thank you for your feedback on our work. Important imaging findings are featured and discussed through the different figures included in the manuscript. Nevertheless, anatomic and pathologic variations of these disease processes require advanced expertise of body/pelvic radiologists. The authors encourage open and ongoing interdisciplinary conversation regarding endometrial cancer findings on a case-by-case basis. |
Reviewer 5 Report
Comments and Suggestions for Authors
The authors present a review style manuscript discussing the role of imaging in endometrial cancer.
The manuscript raises some interesting points however there seems to be a lack of synthesis of ideas through the work.
For instance the manuscript goes into the staging and the histology of endometrial cancer but does quite little to discuss imaging. In fact imaging is simply placed at the end of the manuscript , which seems to be a simple add on to a previous manuscript.
1. What is the main question addressed by the research?
The role of imaging in endo cancer
2. What parts do you consider original or relevant for the field?
It is a review so the reviewers are assessing based on literature/citation comprehension. What specific gap in the field does the paper address? The assumption is that imaging will increase the resolution and accuracy of clinical staging for endometrial cancer.
3. What does it add to the subject area compared with other published
material?
Very little
4. What specific improvements should the authors consider regarding the methodology?
Improvements should include the accurate molecular representation on endometrial cancer. Not simply pole mutations . Seems to be of limited scope. What further controls should be considered?
5. Please describe how the conclusions are or are not consistent with the evidence and arguments presented. Please also indicate if all main questions posed were addressed and by which specific experiments.
Again not fully discussing the molecular heterogeneity of endometrial cancer results in skewed conclusions for the role of imaging in cancer.
6. Are the references appropriate?
Could use additional references
7. Please include any additional comments on the tables and figures and quality of the data.
None
Comments on the Quality of English LanguageThe quality of English is decent , but could be improved through further copy editing
Author Response
|
Comments 1: The authors present a review style manuscript discussing the role of imaging in endometrial cancer. The manuscript raises some interesting points however there seems to be a lack of synthesis of ideas through the work. For instance, the manuscript goes into the staging and the histology of endometrial cancer but does quite little to discuss imaging. In fact, imaging is simply placed at the end of the manuscript, which seems to be a simple add on to a previous manuscript.
1. What is the main question addressed by the research? The role of imaging in endo cancer 2. What parts do you consider original or relevant for the field? It is a review so the reviewers are assessing based on literature/citation comprehension. What specific gap in the field does the paper address? The assumption is that imaging will increase the resolution and accuracy of clinical staging for endometrial cancer. 3. What does it add to the subject area compared with other published material? Very little 4. What specific improvements should the authors consider regarding the methodology? Improvements should include the accurate molecular representation on endometrial cancer. Not simply pole mutations. Seems to be of limited scope. What further controls should be considered? 5. Please describe how the conclusions are or are not consistent with the evidence and arguments presented. Please also indicate if all main questions posed were addressed and by which specific experiments. Again, not fully discussing the molecular heterogeneity of endometrial cancer results in skewed conclusions for the role of imaging in cancer. 6. Are the references appropriate? Could use additional references 7. Please include any additional comments on the tables and figures and quality of the data. None
|
|
Response 1: Thank you for your comments and feedback of our manuscript. The aim of this article was to provide a multidisciplinary discussion on the published FIGO 2023 endometrial cancer updates and the role of radiology in the diagnostic pipeline. Several of the main changes emphasized in this update included evidence-based molecular and histopathological aspects of disease that impact prognosis. Additionally, emerging imaging research continue to provide evidence of the importance of radiology in the diagnosis of endometrial cancer.
|
|
4. Response to Comments on the Quality of English Language |
|
Point 1: The quality of English is decent, but could be improved through further copy editing |
|
Response 1: Thank you for your recommendations regarding further editing to improve the quality of our manuscript. We have used our institutional resources, guided by all the reviewer comments, to very and improve the language quality throughout the article. |
Round 2
Reviewer 5 Report
Comments and Suggestions for Authors
The authors addressed reviewer concerns
Comments on the Quality of English LanguageN/a